# END-TO-END ONE-STEP FLOW MATCHING VIA FLOW FITTING

## ABSTRACT

Diffusion and flow-matching models have demonstrated impressive performance in generating diverse, high-fidelity images by learning transformations from noise to data. However, their reliance on multi-step sampling requires repeated neural network evaluations, leading to high computational cost. We propose FlowFit, a family of generative models that enables high-quality sample generation through both single-phase training and single-step inference. FlowFit learns to approximate the continuous flow trajectory between latent noise $x_0$ and data $x_1$ by fitting a basis of functions parameterized over time $t \in [0, 1]$ during training. At inference time, sampling is performed by simply evaluating the flow only at the terminal time $t = 1$, avoiding iterative denoising or numerical integration. Empirically, FlowFit outperforms prior diffusion-based single-phase training methods achieving superior sample quality.

## 1   INTRODUCTION

In recent years, iterative denoising methods such as diffusion models (23; 10; 24) and flow matching (13; 14) have achieved remarkable success across a wide range of generative modeling tasks, including image synthesis, molecular generation, and audio modeling. These methods define a generative process as the solution to a learned differential equation that progressively transforms simple noise into complex data through a series of small, structured updates. Their strong empirical performance stems from their ability to model complex distributions with stable training dynamics and flexible architectures. However, a key limitation of these approaches lies in their sampling efficiency. Since generation is performed by solving a differential equation, typically through numerical integration, these methods often require hundreds of sequential function evaluations at inference time. This iterative sampling procedure can be computationally expensive, slow, and memory-intensive, limiting their practicality in real-time or resource-constrained settings. In this work, we aim to retain the modeling flexibility and training benefits of diffusion-based methods while enabling efficient **single-step generation**.

While recent work has explored accelerating inference in diffusion models through distillation, these approaches typically follow a two-stage training paradigm. In such methods, a pre-trained diffusion model is first learned through standard iterative training, and then a separate model is trained to mimic its behavior in fewer steps, usually by generating a large synthetic dataset of intermediate trajectories (17; 13) or propagating through a series of teacher and student networks (18; 21). This introduces additional complexity, increases memory requirements, and may limit generalization due to reliance on a fixed teacher.

In contrast, we propose FlowFit, a unified, **end-to-end training** approach that learns to generate samples in one step from the outset. Our method directly parameterizes the entire flow using a basis function expansion, enabling the model to learn global transport trajectories in a compact and structured manner. Once trained, sampling from the model requires only evaluating the learned flow at terminal time $t = 1$, given a source point $x_0 \sim \mu_0$ (see examples in Figure 1).

A key insight motivating our approach is that a smooth transformation, such as a flow trajectory, can be directly fitted using the initial value and all its time derivatives. By fitting the trajectory using a set of basis functions anchored at the initial point, we capture the flow's global behavior and

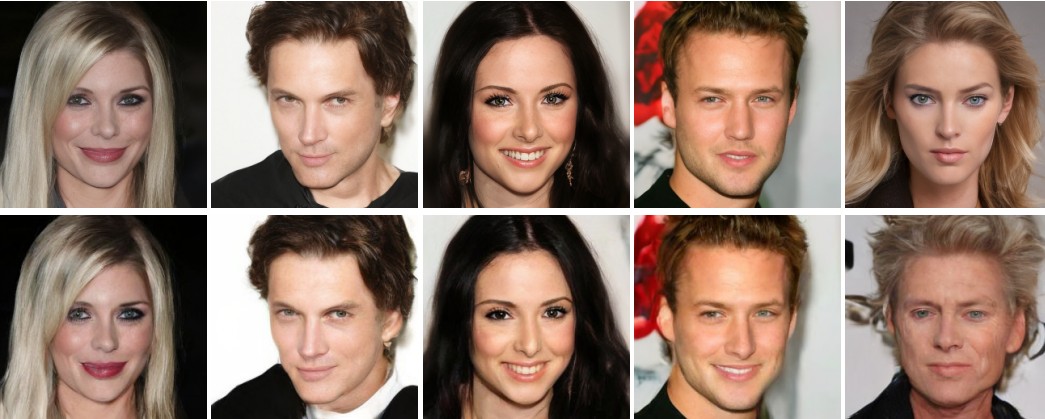

Figure 1: **Comparison Between Multi-Step Flow Matching and Single-Step FlowFit Generation.** The top row shows images generated using a standard flow-matching model with 128 denoising steps, while the bottom row displays outputs from our single-step FlowFit model. Each column uses the same initial noise vector for a fair comparison. FlowFit produces high-fidelity samples even with a single forward pass, offering up to $128\times$ faster sampling than traditional diffusion and flow-matching approaches while maintaining high image quality.

bypass the need for iterative integration. This allows us to retain the expressive modeling capacity of continuous-time methods while achieving orders-of-magnitude faster sampling.

We train the model to ensure that the trajectory is consistent with a learned velocity field through a combination of Conditional Flow Matching and a trajectory-velocity consistency loss. Additionally, we introduce a progressive training strategy that improves stability and alignment between the flow and velocity in early training stages.

Our main contributions can be summarized as follows

- We propose a novel formulation for flow modeling via basis function fitting. FlowFit, a new generative modeling framework that directly parameterizes continuous-time flows using a residual expansion over fixed basis functions. This allows the model to represent entire flow trajectories with a compact, structured parameterization. To the best of our knowledge, FlowFit is the first to propose such a formulation.

- We demonstrate that FlowFit achieves sample quality better or on par with existing single step diffusion-based approaches .

## 2 RELATED WORK

We review prior research efforts aimed at accelerating diffusion-based generative models through single-step sampling. Existing approaches generally fall into two categories: those based on multi-stage distillation and those relying on direct, single-phase training.

### 2.1 TWO-PHASE TRAINING APPROACHES

A common strategy for improving inference efficiency involves distilling multi-step diffusion models into simpler one-step samplers. These techniques typically follow a two-phase process: a full diffusion model is trained first, and then a lightweight student model is optimized to mimic its behavior over fewer denoising steps.

Several works adopt this paradigm by simulating the full denoising trajectory to generate supervision pairs, as seen in knowledge distillation (17) and rectified flows (14). While effective, these methods are computationally intensive due to the need for full reverse-time ODE evaluations. To mitigate this, more recent efforts introduce bootstrapping mechanisms that shorten the ODE simulation path (9; 28).

Additionally, researchers have explored a variety of loss functions beyond the traditional L2 objective, including adversarial criteria (22) and distributional matching techniques (30; 29). Progressive distillation (21; 1; 18) offers a multi-stage solution, wherein a sequence of student models is trained with progressively larger time steps. This hierarchical approach reduces the dependency on costly long-path bootstrap samples. The most similar distillation method to ours is the Physics-Informed Neural Network (27)(PINN). However, there are two key distinctions. First, PINN is a two-stage distillation method that relies on a pre-trained diffusion model, whereas our approach is trained end-to-end in a single stage, eliminating the need for separate distillation. Second, PINN adopts a diffusion-based formulation and uses a single network to jointly model space and time, requiring numerical approximation of flow derivatives, while our method is based on flow matching and decouples time and space via a basis parametrization, allowing exact derivatives of the flow map to be computed analytically and at no extra cost. A concurre

FlowFit diverges from these frameworks by eliminating the need for both pretraining and distillation. Instead, it adopts a unified, end-to-end training scheme that learns a single-step generator directly, simplifying both the implementation and training pipeline.

## 2.2 SINGLE-PHASE TRAINING APPROACHES

Only a limited number of methods have been developed for one-step generation via single-stage training. Among the first of its kind, Consistency Models (26; 25; 16; 3; 8) learn to map noisy inputs directly to their clean counterparts in a single forward pass. Although originally designed for distillation, they have also been extended to an end-to-end training setup. iCT (25) and sCT (16) refine Consistency Models by altering the training procedure, leading to gains in sample quality and stability. sLST (3) and ECT (8) propose enhanced optimization strategies for consistency models including an improved loss formulation, training schedules and regularization techniques to stabilize learning and significantly boost sample quality while maintaining fast, few-step generation. In contrast, Shortcut Models (6) formulate generation as a process conditioned jointly on the current noise level and the chosen step size, which makes the method adaptable to different inference-time compute budgets. Shortcut models (6) introduce a flexible generative approach that allows conditioning on both the input noise level and the desired step size, enabling inference under various computational constraints. IMM (31) is a training-based single-step generative method that matches low-order moments of the target transition distributions at each step. By focusing on these statistics, it enables efficient one or few-step sampling without relying on pre-trained models or multi-stage distillation.

A concurrent work, Mean Flows (7), proposes a one-step generative modeling framework that learns the average flow along the trajectory. In contrast, our approach is orthogonal: we explicitly fit the flow using a basis of functions, enabling a flexible and exact representation of the full generative trajectory.

To the best of our knowledge, FlowFit is the first method to directly aim for a single-step generative model using a basis of functions.

## 3 PRELIMINARY: DIFFUSION AND FLOW MATCHING

Recent advances in generative modeling have led to the development of methods such as diffusion models (23; 10; 24) and flow-matching approaches (13; 14), which learn a continuous-time transformation from a simple noise distribution to a complex data distribution. These models typically define the generative process through an ordinary differential equation (ODE), where the time-dependent dynamics are learned to guide samples from noise toward data.

In this work, we adopt the flow-matching formulation based on the optimal transport objective introduced by (14), as it offers a simple and effective framework for learning such dynamics. While diffusion models and flow-matching approaches are often studied separately, recent perspectives, such as that of (12), highlight that flow matching can be interpreted as a deterministic special case of diffusion modeling. Accordingly, we treat the two paradigms as closely related and use the terminology interchangeably where appropriate.

Flow Matching provides a supervised learning framework for modeling deterministic, continuous-time flows that transport a base distribution $\mu_0$ (e.g., standard Gaussian) into a target distribution $\mu_1$

(e.g., data distribution). Drawing from optimal transport and neural ODEs, it directly learns a velocity field that defines a transport trajectory between paired samples.

Let $x_0 \sim \mu_0$ be a sample from the source distribution and $x_1 \sim \mu_1$ its corresponding target. The model learns a time-dependent velocity field $\mathbf{v}_\theta(x, t)$ such that solving the associated ODE transforms $\mu_0$ into $\mu_1$. Formally, the flow is described by

$$\begin{cases} \dfrac{d}{dt}\psi_t(x) = \mathbf{u}_t(\psi_t(x)), \\ \quad \psi_0(x) = x, \end{cases} \tag{1}$$

where $\mathbf{u}_t : [0, 1] \times \mathbb{R}^d \to \mathbb{R}^d$ is a neural network parameterizing the velocity field, and $\psi_t(x)$ is the flow map at time $t$.

A common training strategy is to supervise the model using velocity information along linear paths between $x_0$ and $x_1$, evaluated at intermediate points $(1 - t)x_0 + tx_1$. The corresponding ground-truth velocity at such a point is simply $x_1 - x_0$.

The model is then trained to match this known velocity at intermediate points. The Conditional Flow Matching (CFM) objective function is defined as

$$\mathcal{L}(\theta) = \mathbb{E}_{x_0 \sim \mu_0, \, x_1 \sim \mu_1, \, t \sim \mathcal{U}[0,1]} \left[ \left\| \mathbf{v}_\theta \left( (1 - t)x_0 + tx_1, t \right) - (x_1 - x_0) \right\|^2 \right]. \tag{2}$$

This loss guides the model to predict the instantaneous velocity field that aligns with the linear flow between samples. Notably, this avoids the need for computing density functions or score gradients, distinguishing it from traditional diffusion models.

Once training is complete, sample generation begins by drawing an initial point $x_0 \sim \mu_0$, typically from a standard Gaussian distribution. This point is then transformed toward the data distribution by solving the learned ODE defined by the velocity field $\mathbf{v}_\theta(x, t)$. In practice, this continuous-time flow is discretized and approximated using numerical integration methods such as Euler's method, where the sample is updated iteratively over a sequence of small time steps from $t = 0$ to $t = 1$.

# 4 FLOWFIT: DIRECT FLOW PARAMETERIZATION VIA BASIS FUNCTION FITTING

In this section, we introduce FlowFit, a novel approach for modeling continuous-time flows via basis function fitting. Specifically, we aim to directly parameterize the flow $\psi_t(x)$ using a basis of functions that are conditioned on both the initial point $x_0$ and the time parameter $t$. Thus, the goal is to model the mapping $(x_0, t) \mapsto \psi_t(x) = x_t$, where $\psi_t(x)$ is the flow at time $t$, and $x_t$ represents the transformed point at time $t$.

To this end, we approximate the true flow $\psi_t(x)$ with a learnable function $\psi_\theta(x_0, t)$[1], parameterized via neural networks and basis functions. For simplicity, we use the notation $x_t$ interchangeably with both $\psi_t(x)$ and $\psi_\theta(x_0, t)$, with the dependence on $x_0$ and $t$ understood implicitly. This allows us to define the transformation in terms of a set of basis functions, which we leverage to approximate the flow dynamics. Importantly, this representation enables efficient single-step generation by evaluating the learned trajectory at $t = 1$, and requires only the initial sample $x_0 \sim \mu_0$ at inference time.

## 4.1 TRAJECTORY PARAMETERIZATION

We approximate the continuous flow trajectory using the time-dependent formulation

$$\begin{cases} \psi_\theta(x_0, t) & = x_0 + \sum_{k=1}^{K} f_{\theta,k}(x_0)(\gamma_k(t) - \gamma_k(0)), \\ \psi_\theta(x_0, 0) & = x_0, \end{cases} \tag{3}$$

where

---

[1] In the general case, we aim to learn $\psi_\theta(x_0, c, t)$, where $c$ denotes any form of conditioning (e.g., class labels). For simplicity of derivation, we omit $c$.

- $\{\gamma_k(t)\}_{k=1}^{K}$ are fixed scalar basis functions (e.g., polynomial or Fourier),
- $f_{\theta,k} : \mathbb{R}^d \to \mathbb{R}^d$ are neural networks that produce the coefficients,
- $\theta$ denotes all learnable parameters.

We note that this parametrization satisfies the boundary condition $\psi_\theta(x_0, t \approx 0) \approx x_0$, *by construction*. A natural question is whether the trajectory $\psi(x, t)$ can be approximated arbitrarily well using the basis expansion above.

## 4.2 THEORETICAL JUSTIFICATION FOR BASIS FUNCTION FLOW MODELING

A key question in FlowFit is whether the proposed basis expansion reliably approximates the target flow. The following proposition confirms that this is indeed the case.

**Proposition 1** (Universal Approximation of Flow Trajectories Using a Basis of Functions). *Let $\psi : (\mathbb{R}^d \times [0, 1]) \to \mathbb{R}^d$ be a continuous trajectory from an initial point $x_0$ to a target point $x_1$. Then, for any $\varepsilon > 0$, there exists a sufficiently large integer $N > 0$, a set of basis functions $\{\gamma(t)\}_{i=1}^{N}$, and coefficients $\{W_i(x)\}_{i=1}^{N}$ such that*

$$\left\| \psi(x, t) - \left( \sum_{i=1}^{N} \gamma_i(t) \cdot W_i(x) \right) \right\| < \varepsilon, \quad \forall t \in [0, 1].$$

We include a proof in the appendix A for completeness.

## 4.3 FLOWFIT TRAINING

We jointly train two models: $\psi_\theta$, which parameterizes the flow trajectory, and $v_{\theta'}$, which models the time-dependent velocity field. To ensure that the learned trajectory $\psi_\theta(x_0, t)$ evolves in alignment with the correct transport dynamics, we supervise it by matching its time derivative to a known target velocity field. Taking the derivative of the basis function parameterization gives

$$\frac{d\psi_\theta}{dt}(x_0, t) = \sum_{k=1}^{K} f_{\theta,k}(x_0) \frac{d\gamma_k}{dt}(t). \tag{4}$$

Simultaneously, the velocity model $v_{\theta'}(x, t)$ is trained using the standard Conditional Flow Matching (CFM) objective. In this setup, a source point $x_0$ is randomly sampled from the base distribution $\mu_0$ and a target point $x_1$ is independently sampled from the data distribution $\mu_1$. The model is then trained to match the ground-truth velocity $\tilde{\mathbf{v}}(x_0, x_1, t) = x_1 - x_0$ at intermediate points along the linear interpolation between $x_0$ and $x_1$. The loss is given by

$$\mathcal{L}_{\text{CFM}}(\theta') = \mathbb{E}_{x_0 \sim \mu_0, \, x_1 \sim \mu_1, \, t \sim \mathcal{U}[0,1]} \left[ \| v_{\theta'}((1-t)x_0 + tx_1, t) - (x_1 - x_0) \|^2 \right]. \tag{5}$$

To align the basis-induced flow with the learned velocity field, we require that the velocity induced by the basis-function trajectory matches the prediction of the velocity model at the corresponding location along the flow

$$\frac{d\psi_\theta}{dt}(x_0, t) \approx v_{\theta'}(\psi_\theta(x_0, t), t). \tag{6}$$

We formalize this requirement with a loss that enforces matching the flow derivative to the velocity

$$\mathcal{L}_{\text{derivative}}(\theta) = \mathbb{E}_{x_0 \sim \mu_0, \, t \sim \mathcal{U}[0,1]} \left[ \left\| \frac{d\psi_\theta}{dt}(x_0, t) - v_{\theta'}(\text{sg}[\psi_\theta(x_0, t)], t) \right\|^2 \right], \tag{7}$$

where $\text{sg}[\cdot]$ denotes the stop-gradient. This loss encourages the parameterized trajectory to follow a velocity field that is internally consistent with the learned dynamics, improving the alignment between the path and the underlying transport vector field.

Despite the fact that $\psi_\theta(x_0, t)$ may initially provide limited information early in training, we observe that the propagation loss remains robust and effective. This eliminates the need for manually

---

**Algorithm 1** Training FlowFit

1: Initialize $\theta, \theta'$, time window $\alpha_t \leftarrow 0$
2: Initialize $\psi_\theta(., t)$ as in Equation 3
3: **for** each training step **do**
4:     Sample $x_0 \sim \mu_0$, $x_1 \sim \mu_1$, $t \sim \mathcal{U}[0, 1]$
5:     **Train $v_{\theta'}$ with CFM:**
6:       $\tilde{x}_t = (1 - t)x_0 + tx_1$
7:       $\min_{\theta'} \mathcal{L}_{\text{CFM}}(\theta') = \|v_{\theta'}(\tilde{x}_t, t) - (x_1 - x_0)\|^2$
8:     Update $\theta'$
9:     **Train $\psi_\theta$ with consistency loss:**
10:     Compute $\psi_\theta(x_0, t)$, $\frac{d\psi_\theta}{dt}(x_0, t)$ using Equation 4
11:     $\min_\theta \mathcal{L}_{\text{derivative}}(\theta) = \|\frac{d\psi_\theta}{dt}(x_0, t) - v_{\theta'}(\psi_\theta(x_0, t), t)\|^2$
12:     Update $\theta$
13:     Increase $\alpha_t$ toward 1
14: **end for**

---

**Algorithm 2** Sampling

1: Sample $x_0 \sim \mu_0$
2: **Return** $x_1 = \psi_\theta(x_0, t = 1)$

---

scheduling $t$, thereby simplifying the training process and enhancing both stability and usability, all without sacrificing performance.

We emphasize that training the velocity $v_{\theta'}$ is independent of training $\psi_\theta$, allowing both to be trained *jointly* and *in parallel*. Consequently, the effective training time nearly matches that of a single generative model.

The full training algorithm and the corresponding sampling procedure are outlined in Algorithm 1 and 2.

### 4.4 SINGLE-STEP GENERATION

At inference time, the model generates samples from the target distribution $\mu_1$ by drawing $x_0 \sim \mu_0$ and evaluating the fitted trajectory at terminal time:

$$x_1 = \psi_\theta(x_0, t = 1). \tag{8}$$

This enables fast and deterministic generation without iterative integration, in contrast to diffusion or traditional flow-based models.

## 5 EXPERIMENTS

### 5.1 EXPERIMENTAL SETUP

We evaluate our method alongside a range of established baselines under consistent training conditions. To ensure fairness, all models are trained from scratch using an identical implementation and share the same backbone architecture, the DiT-B diffusion transformer (19). Our evaluation includes two tasks: unconditional image generation on the CelebAHQ-256 dataset (15) and a comparison with class-conditional generation on ImageNet-256 (5). For ImageNet experiments, we use the classifier-free guidance (CFG) (11) is employed to enhance conditional generation. For the experiments reported in Table 6, we use the AdamW optimizer. We use a Polynomial basis with order 8. All models are trained and sampled in the latent space provided by the `sd-vae-ft-mse` autoencoder (20). Further implementation details are provided in Appendix C. We release the full code in the supplementary materials.

### 5.2 BASELINE APPROACHES

For comparison, we consider several end-to-end generative modeling approaches under the evaluation protocol of (6). Consistency Models (26) learn one-step generation by training on empirical pairs $(x_t, x_{t+\delta})$, with time discretization granularity refined progressively during training. Improved

Table 1: **Comparison of various training objectives applied to the same architecture (DiT-B)**. We report FID-50k scores (lower is better) for 128, 4, and 1-step denoising. FlowFit achieves high-quality samples using a single training phase and a one-step inference process. Results in parentheses indicate settings beyond the intended use of the corresponding objective.

| Single-stage methods | CelebAHQ-256 | | | ImageNet-256 (Class-Conditional) | | |
|---|---|---|---|---|---|---|
| | **128-Step** | **4-Step** | **1-Step** | **128-Step** | **4-Step** | **1-Step** |
| Diffusion (10) | 23.0 | (123.4) | (132.2) | 39.7 | (464.5) | (467.2) |
| Flow Matching (13) | 7.3 | (63.3) | (280.5) | 17.3 | (108.2) | (324.8) |
| CT (26) | 53.7 | 19.0 | 33.2 | 42.8 | 43.0 | 69.7 |
| iCT (25) | - | - | 21.7 | - | - | 43.3 |
| sCT (16) | - | - | 19.3 | - | - | 41.6 |
| Shortcut Models (6) | **6.9** | **13.8** | 20.5 | **15.5** | **28.3** | 40.3 |
| IMM (31) | - | - | 19.5 | - | - | 41.4 |
| sLST (3) | - | - | 18.8 | - | - | 39.9 |
| ECT (8) | - | - | 20.7 | - | - | 40.6 |
| FlowFit (ours) | - | - | **14.1** | - | - | **34.4** |

variants such as iCT (25) and sCT (16) modify the optimization strategy to enhance training stability and overall sample quality. Shortcut Models (6) instead condition the generator jointly on the current noise level and the chosen step size, allowing adaptive sampling under different computational budgets. Another relevant approach is Live Reflow (6), which jointly trains on both flow-matching objectives and distillation-based targets. However, because it requires generating new targets via full denoising at every iteration, this method incurs significant computational overhead.

## 5.3 EVALUATION PROTOCOL

We follow the evaluation framework established in (6). Each model generates 50k samples for computing the FID-50k score. Our method is evaluated using a single-step sampler, while the baseline models are assessed under 128-step, 4-step, and 1-step variants. FID-50k is calculated using statistics from the full dataset, with no compression applied to the generated samples. All images are resized to $299 \times 299$ via bilinear interpolation and normalized to the $[-1, 1]$ range. During inference, we apply the Exponential Moving Average (EMA) of the model parameters to improve stability and performance.

## 5.4 COMPARISON

Table 6 highlights that FlowFit delivers high-quality generations in the single sampling step. Notably, it surpasses all other single-phase training methods in one-step generation performance. Figure 3 show example generations at 256×256 resolution on CelebA-HQ. Additional qualitative results are presented in Appendix B.

## 5.5 SEMANTIC STRUCTURE IN THE LATENT SPACE OF FLOWFIT

To assess whether FlowFit gives rise to a semantically meaningful and smooth latent space, we perform an interpolation experiment in the input noise domain. We begin by selecting pairs of Gaussian noise vectors $x_0^0$ and $x_0^1$, and interpolate between them using a variance-preserving scheme $x_0^n = n x_0^1 + \sqrt{1 - n^2}\, x_0^0$ with $n \in [0, 1]$. Each interpolated point $x_0^n$ is then processed through the trained model to generate the corresponding output. Figure 2 shows representative results from this interpolation. Even though no explicit smoothness constraints or regularization terms are imposed during training, the outputs exhibit continuous and visually coherent changes. The interpolated generations preserve high-level semantics while gradually morphing between endpoints, indicating that FlowFit captures an underlying latent structure that supports semantically consistent transitions.

Figure 2: **Latent Space Interpolation.** All images shown are generated by the model. Each row illustrates the result of applying one-step denoising to intermediate samples obtained by variance-preserving interpolation between two independent Gaussian noise vectors.

Table 2: Impact of the chosen polynomial basis order on image quality when training FlowFit on CelebAHQ-256 using a batch size of 64 and 400,000 training iterations..

| Basis Order | $\{t^k\}_{k=1}^2$ | $\{t^k\}_{k=1}^4$ | $\{t^k\}_{k=1}^8$ |
|---|---|---|---|
| FID ($\downarrow$) | 18.2 | 15.9 | 14.1 |

# 6 ABLATIONS

## 6.1 EFFECT OF BASIS ORDER

We investigate how the order of the polynomial basis affects image quality. Table 2 reports results for different basis orders under identical training conditions: order 2 with basis functions $\{t^k\}_{k=1}^2$, order 4 with basis functions $\{t^k\}_{k=1}^4$, and order 8 with basis functions $\{t^k\}_{k=1}^8$. Our results indicate that higher-order bases consistently yield improved image quality.

Table 5: Effect of the DiT backbone architecture on image quality when training FlowFit on CelebAHQ-256 using a batch size of 64 and 400,000 training iterations.

| | DiT-B | DiT-L | DiT-XL |
|---|---|---|---|
| FID | 14.1 | 10.7 | 6.2 |

## 6.2 EFFECT OF THE BASIS TYPE

We evaluate the influence of the basis function type on generation quality while keeping the expansion order fixed. Specifically, we use $\{t^k\}_{k=1}^8$ for the polynomial basis and $\{\cos(2k\pi t)\}_{k=1}^4 \cup \{\sin(2k\pi t)\}_{k=1}^4$ for the trigonometric basis. As reported in Table 3, the polynomial basis yields a better FID score of 14.1 compared to the trigonometric basis (16.3).

## 6.3 EFFECT OF STOP GRADIENT IN EQUATION 7) DURING TRAINING

In Table 4, we report the FID scores of generated images when training with and without gradient stopping. We observe that disabling gradient stopping results in an additional computational overhead and lower image quality.

## 6.4 EFFECT OF THE NETWORK BACKBONE

Table 5 reports the effect of the DiT backbone architecture on image quality when training FlowFit on CelebA-HQ. We observe that increasing the model capacity consistently improves performance: using DiT-L instead of DiT-B reduces the FID from 14.1 to 10.7, while DiT-XL further lowers it to 6.2. These results demonstrate that FlowFit benefits from larger backbones, achieving higher-quality samples with increased model capacity.

Table 3: Impact of the basis nature using the same expansion order.

| Basis | Polynomial | Trigonometric |
|---|---|---|
| FID ↓ | 14.1 | 16.3 |

Table 4: Effect of applying gradient stopping at training (as in Equation 7) for CelebAHQ-256.

| | w/o stopping gradient | w/ stopping gradient |
|---|---|---|
| FID ↓ | 52.4 | 14.1 |

## 7 LIMITATIONS AND FUTURE WORK

While FlowFit demonstrates strong performance, it has certain limitations. A key drawback is that the current design is restricted to single-step inference. Another potential improvement would be to use a single network instead of two during training. A promising direction for future research is to generalize the framework to a unified model architecture that supports flexible sampling with a variable number of inference steps.

Due to computational constraints, we explore the method up to order 8 in this work. It would be interesting to investigate higher-order expansions and assess their impact. Another potential direction is to explore alternative basis functions in the formulation.

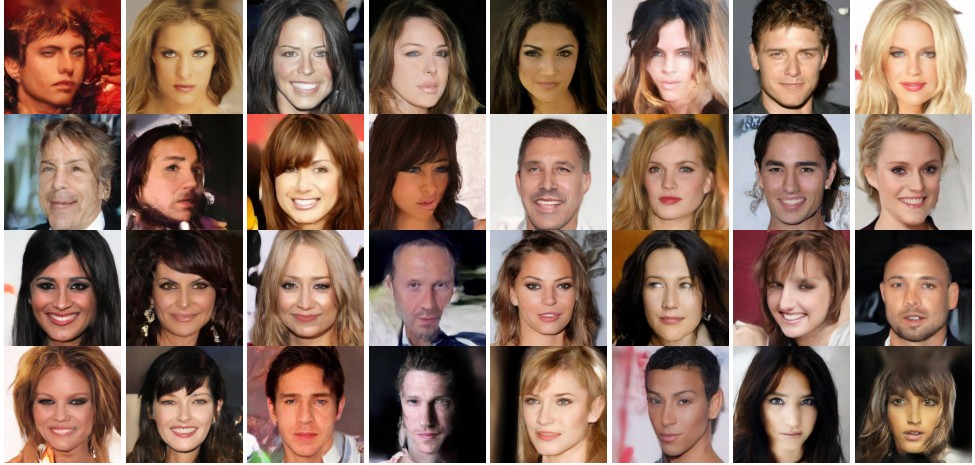

Figure 3: Unfiltered samples generated using FlowFit on the unconditional CelebA-HQ dataset at a resolution of 256×256. These images were produced in a single forward pass using a DiT-B model trained for 400,000 iterations.

## 8 REPRODUCIBILITY STATEMENT

To support reproducibility, we provide a complete implementation of our method, including training and evaluation scripts, as part of the supplementary materials.

## 9 CONCLUSION

We introduce FlowFit, a novel generative model that enables single-step sampling. The key idea is a new formulation of flow modeling through basis function fitting, which allows the model to learn the generative trajectory efficiently. As a result, FlowFit achieves fast, high-quality generation making it a practical solution for single-step generative modeling.

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

Table 6: **Comparison of the distillation version of our method with some sample distillation methodswhen using the same architecture (DiT-B)**. We report FID-50k scores (lower is better) for 128, 4, and 1-step denoising.

| Distillation methods | CelebAHQ-256 | | | ImageNet-256 (Class-Conditional) | | |
|---|---|---|---|---|---|---|
| | 128-Step | 4-Step | 1-Step | 128-Step | 4-Step | 1-Step |
| Progressive Distillation | (302.9) | (251.3) | **14.8** | (201.9) | (142.5) | 35.6 |
| Consistency Distillation | 59.5 | 39.6 | 38.2 | 132.8 | 98.01 | 136.5 |
| Reflow | **16.1** | **18.4** | 23.2 | **16.9** | **32.8** | 44.8 |
| Ours (Distillation) | - | - | **13.4** | - | - | **31.6** |

## A  THEORETICAL JUSTIFICATION FOR BASIS FUNCTION FLOW MODELING

A key question in FlowFit is whether the proposed basis expansion reliably approximates the target flow. The following proposition confirms that this is indeed the case.

**Proposition 2** (Universal Approximation of Flow Trajectories). *Let $\psi : (\mathbb{R}^d \times [0,1]) \to \mathbb{R}^d$ be a continuous trajectory from an initial point $x_0$ to a target point $x_1$. Then, for any $\varepsilon > 0$, there exists a sufficiently large integer $N > 0$, a set of basis functions $\{\gamma(t)\}_{i=1}^N$, and coefficients $\{W_i(x)\}_{i=1}^N$ such that*

$$\left\| \psi(x,t) - \left( \sum_{i=1}^N \gamma_i(t) \cdot W_i(x) \right) \right\| < \varepsilon, \quad \forall t \in [0,1].$$

*Proof.* The results is immediately obtained by applying the Stone–Weierstrass theorem (4; 2) because $\mathbb{R}^d$ and $[0,1]$ are both locally compact Hausdorff spaces, and the basis functions and the coefficients are all continuous. $\square$

## B  QUALITATIVE SAMPLES

Figures 4 and 5 present sample outputs from models trained on CelebA-HQ (unconditional) and ImageNet (class-conditioned), respectively, using our proposed training procedure.

## C  IMPLEMENTATION DETAILS

## D  TRAINING DETAILS

Table 7 provides detailed training configurations corresponding to the results reported in Table 1 (main paper) Across all experiments, we train models using a latent representation obtained from the `sd-vae-mse-ft` encoder with a downsampling factor of 8, mapping $(256 \times 256 \times 3)$ images to a $(32 \times 32 \times 4)$ latent space. For CelebA-HQ, we use a batch size of 64 and train for $400k$ steps, while for ImageNet we use a batch size of 256 and train for $800k$ steps. The ImageNet model additionally employs classifier-free guidance with a scale of 1.5 and a class-dropout probability of 0.1, whereas the CelebA-HQ model is trained without guidance. We use AdamW with a learning rate of $(5 \times 10^{-5})$, zero weight decay, and maintain an exponential moving average of parameters with decay 0.999 for evaluation. The backbone architecture uses a hidden size of 768, patch size of 2, 12 layers, 12 attention heads, and an MLP expansion ratio of 4. All experiments use a progressive step size of $(\delta t = 0.01)$.

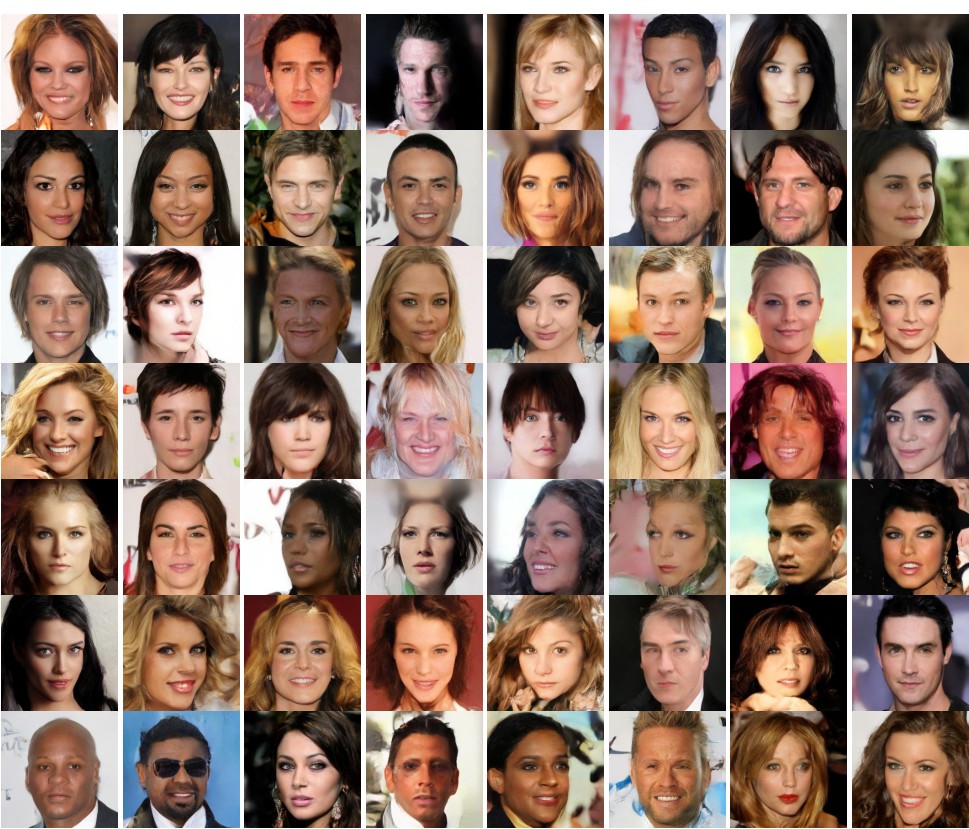

Figure 4: Unfiltered samples generated on the unconditional CelebA-HQ dataset at a resolution of 256×256. These images were produced in a single forward pass using a DiT-B model trained for 400,000 iterations.

702
703
704
705
706
707
708
709
710
711
712
713
714
715
716
717
718
719
720
721
722
723
724
725
726
727
728
729
730
731
732
733
734
735
736
737
738
739
740
741
742
743
744
745
746
747
748
749
750
751
752
753
754
755

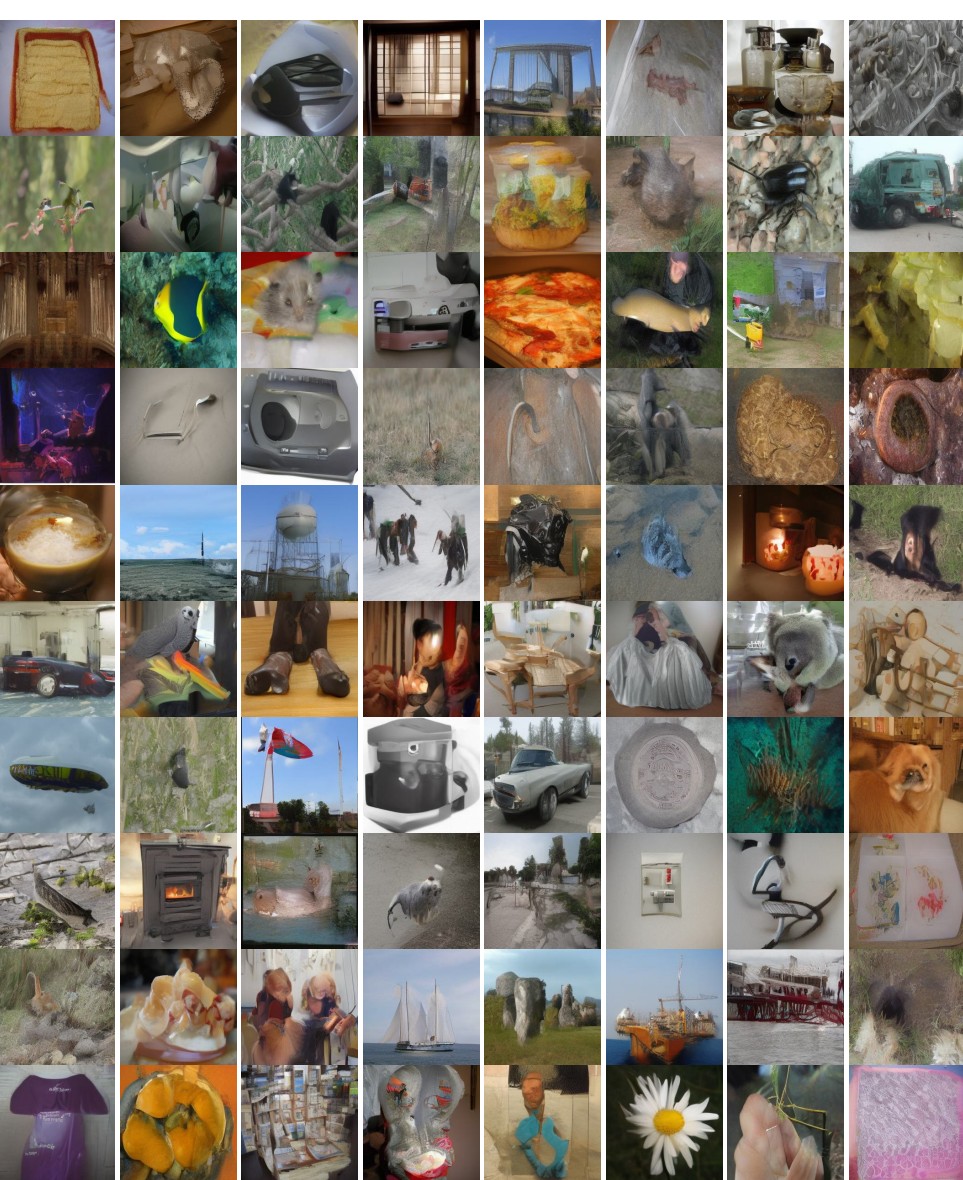

Figure 5: Unfiltered samples generated on the unconditional ImageNet dataset at a resolution of 256×256. These images were produced in a single forward pass using a DiT-B model trained for 800,000 iterations.

| | |
|---|---|
| Batch Size | 64 (CelebA-HQ), 256 (Imagenet) |
| Training Steps | 400,000 (CelebA-HQ), 800,000 (Imagenet) |
| Latent Encoder | sd-vae-mse-ft |
| Latent Downsampling | 8 (256x256x3 to 32x32x4) |
| Classifier Free Guidance | 0 (CelebA-HQ), 1.5 (Imagenet) |
| Class Dropout Probability | 0 (CelebA-HQ), 0.1 (Imagenet) |
| EMA Parameters Used For Evaluation? | Yes |
| EMA Ratio | 0.999 |
| Optimizer | AdamW |
| Learning Rate | 0.00004 |
| Weight Decay | 0.0 |
| Hidden Size | 768 |
| Patch Size | 2 |
| Number of Layers | 12 |
| Attention Heads | 12 |
| MLP Hidden Size Ratio | 4 |
| Basis | Polynomial |
| Basis order | 8 |

Table 7: Default hyperparameter settings used during training.

