# OpenReview forum: "End-to-End One Step Flow Matching via Flow Fitting"
_ICLR.cc/2026/Conference — Submitted to ICLR 2026_

### Official Review · Reviewer_vFLX · 2025-10-29

**Soundness:** 2
**Presentation:** 3
**Contribution:** 2
**Rating:** 4
**Confidence:** 3

**Summary:**

The paper proposes a single-phase method to learn one-step map to approximate the ODE trajectory in flow matching to enable one-step generation. The authors introduce basis functions in time to model the one-step map to reduce the computation of backpropagation in training.

**Strengths:**

The paper is well organized and easy to follow. The authors introduced fixed basis functions in time to model the one-step map in order to reduce the computation of backpropagation in training. Experiment results also verify the efficiency of the proposed method.

**Weaknesses:**

The idea is quite naive overall, using a one-step map to approximate the ODE trajectory via matching velocity. Although the author introduces the basis functions, it is basically doing distillation over a pre-trained flow matching model. I understand that this is a single phase method and the training of one-step map can be in parallel with FM, but the computation cost is still doubled. The method is not very appealing to me.

**Questions:**

Can the authors provide results for multiple step evaluation of the proposed method? I understand the primary goal is to do one-step generation, but I still want to know if the proposed method can work well when increasing computational budget.

---

> ### Author Response · Authors · 2025-12-03
>
> Thank you for your feedback. Please see the detailed response below.
>
> > The idea is quite naive overall, using a one-step map to approximate the ODE trajectory via matching velocity.
>
> We respectfully disagree. Our key contribution is the introduction of **basis-function fitting** within the context of 1-step diffusion and flow matching, an approach that, to the best of our knowledge, has not been explored before and is neither trivial nor naive. Moreover, we argue that the simplicity of our method is **not a handicap**, rather, it is an **advantage**, enabling straightforward implementation and broader usability.
>
> > Although the author introduces the basis functions, it is basically doing distillation over a pre-trained flow matching model. I understand that this is a single phase method and the training of one-step map can be in parallel with FM, but the computation cost is still doubled.
>
> To address the concern regarding training cost, we emphasize that what ultimately matters is the **number of forward and backward passes per iteration** compared to **existing end-to-end 1-step diffusion and flow-matching methods**. The table below summarizes this comparison.
>
> | Method | Number of forward  passes per iteration  |Number  of backward passes per iteration |
> |:------:|------------------------|---------------------------|
> | Consistency Models [26, 23, 24] | 2  | 1 |
> | Shortcut Models [5]    | 3 | 1 |
> | Ours  |3 | 2 |
>
> Our method introduces a slight computational overhead compared to state-of-the-art Shortcut Models. However, since we train **two separate networks**, some of the forward and backward passes can be executed **in parallel**, reducing the effective sequential cost. In practice, our total compute is only marginally higher than Shortcut Models.
>
> Shortcut Models compress both the velocity predictor and the shortcut predictor into a single network $f(x_t, t, d)$, where $(d = 0)$ outputs the velocity and $(0 < d \le 1$) outputs the shortcut prediction. This forces one architecture to represent two functions simultaneously, creating a trade-off in representational capacity and accuracy. In contrast, our method uses two specialized networks, each dedicated to a distinct role, and fully parallelizable during training. This design increases effective capacity without increasing sequential training cost. If desired, we believe that our formulation can also be implemented using a single unified network (similar to Shortcut Models) with an additional input flag, while still preserving our basis-function structure. We chose the two-network design because it enables parallel training and provides greater modeling flexibility.
>
> > Can the authors provide results for multiple step evaluation of the proposed method? I understand the primary goal is to do one-step generation, but I still want to know if the proposed method can work well when increasing computational budget.
>
> Although our method is mainly proposed for the 1-step setting, it can still enable few-step inference. One option is to combine our learned 1-step map (produced by the second network) with the standard velocity model (learned by the first network).
> The idea is to map the initial Gaussian noise to an intermediate state $x_{t_i}$ that corresponds to an intermediate tilmestep $t_i$ in a single step using our 1-step model, and then perform (N-1) additional steps using a standard ODE integration scheme applied to the velocity model starting from $x_{t_i}$. In the table below, we report results on CelebA-HQ. We observe a slight improvement in FID when adding more steps. We chose $t_i = 1 - 0.02 \times \text{number of steps}$.
>
> | Method | 1-step | 2-step | 4-step | 8-step |
> | ------ | ------ | ------ | ------ | ------ |
> | Ours   | 14.1   | 13.8   | 12.9   | 12.1   |

---

### Official Review · Reviewer_YPvN · 2025-10-31

**Soundness:** 2
**Presentation:** 2
**Contribution:** 2
**Rating:** 4
**Confidence:** 4

**Summary:**

This paper presents FlowFit, a novel framework for flow modeling via basis function fitting that enables high-quality sample generation through both single-phase training and single-step inference, which avoids iterative numerical integration like denoising diffusion models. To do this, the paper directly parameterizes the continuous-time flows using a residual expansion over fixed basis functions. This framework can be used in one step generation and distillation of pretrained models. The paper experiments on CelebAHQ-256 and ImageNet-256 to show that the proposed method outperforms prior diffusion-based single-phase training methods.

**Strengths:**

•	The paper is clearly organized with direct motives.

•	The proposed method of using basis function to directly distill the learned velocity field is simple and straightforward.

**Weaknesses:**

•	The content of the paper is inadequate. It would be beneficial to add more discussions on the choice of basis functions and the capacity on approximating certain flows for specific tasks. It remains suspicious to me that the proposed method could scale up to more general complex generation tasks, in which the optimal velocity field and the flow map is highly curved and the degree of the mapping function is large.

•	The paper lacks theoretical analysis of the proposed method. It would be beneficial to theoretically discover how large the basis function family is needed for a given task in mathematical details, a simple synthetic example could suffice. The comparison may utilize the theory in polynomial regression analysis.

•	In my view, the empirical comparison in the experiments is a bit obscure and misleading. It is somewhat unfair to put the proposed method FlowFit in the column of “1-step” with the other method’s “1-step” like standard flow matching and consistency models, since the FlowFit method regards the multi-step trajectory as a single complex function map. The complexity of the “1-step” for FlowFit method is much higher than the “1-step” for a single step of sample update according to the learned velocity field in standard flow matching and consistency models. It would be more convincing to show the effectiveness of the proposed method by presenting the exact numerical comparison of the total time cost.

**Questions:**

•	Besides the empirical comparison, could the authors provide an understanding on why directly fitting the map could be better than other few step generation methods based on updating samples according to the velocity field? It seems to me that the difficulty in fitting the velocity field is not less than directly updating the samples through the distilled velocity field in previous few step generation methods like diffusion distillation.

•	Could the flow fitting strategy be applied to other classical flow trajectories like diffusion paths in one step generation tasks like diffusion distillation? If yes, will the type of the true flow trajectory affect the choice of the basis function family and the difficulty of flow fitting?  Could the authors provide some understanding on whether and why the flow fitting strategy will outperform other diffusion distillation methods in this case?

---

> ### Author Response · Authors · 2025-12-03
>
> Thank you for your feedback. Please see the detailed response below.
>
> > It would be beneficial to add more discussions on the choice of basis functions and the capacity on approximating certain flows for specific tasks. It remains suspicious to me that the proposed method could scale up to more general complex generation tasks, in which the optimal velocity field and the flow map is highly curved and the degree of the mapping function is large.
>
> Regarding the theoretical existence of a basis capable of fitting the flow for a given task or dataset, we note that in Lines 217–219, we explicitly raise the question of whether the flow trajectory can be effectively approximated using a basis of functions. To support this, we include in the supplementary material the general **Stone–Weierstrass theorem**, which states that on a compact domain ($t \in [0, 1]$), there exists a set of continuous functions capable of approximating any continuous function. In particular, the **Weierstrass approximation theorem** ensures that any continuous function defined on a compact space can be approximated arbitrarily well by polynomials. We believe this provides a solid theoretical foundation for the existence of such approximations for **any task or dataset**. We moved the theoretical analysis from the supplementary to the main paper.
>
> > It would be beneficial to theoretically discover how large the basis function family is needed for a given task in mathematical details.
>
> By analogy with deep neural networks, the **order of the basis** (i.e., its size) is similar to the **depth of a network**. The universal approximation theorem states that even a single hidden layer can approximate any continuous function, but empirically, adding more layers often improves performance. Similarly, in our case, we observe that performance improves and **does not decrease** as the order of the polynomial basis increases (see Table 2 in the main paper).
>
> > It would be more convincing to show the effectiveness of the proposed method by presenting the exact numerical comparison of the total time cost.
>
> | Method                                     | Inference Time (ms) | FID   |
> |-------------------------------------------|-------------------|-------|
> | iCT [23]                                   | 64                | 21.4  |
> | sCT [14]                                   | 64                | 19.2  |
> | Shortcut Models [5]                        | 64                | 20.4  |
> | Ours (multi-process, basis order = 8)     | 68                | **14.1**  |
> | Ours (basis order = 2)                     | 82                | 18.2  |
> | Ours (basis order = 4)                     | 110               | 15.9  |
> | Ours (basis order = 8)                     | 161               | **14.1**  |
>
>
> Here we provide a **numerical comparison of the total time cost** in case of 1-step inference and CelebAHQ dataset. When scaling compute resources, our inference can match existing methods, as all basis coefficients can be computed in parallel in separate processes (Ours (multi-process)), resulting in similar latency. In the table above, we also report the latency when basis coefficients are computed **only along the batch dimension**, which introduces additional overhead as the basis order increases.  Our method leverages the higher capacity of the basis, resulting in **higher image quality**. Notably, even with **basis order = 2**, our method achieves better FID than the Shortcut Models while incurring only a **small increase in inference time**.
>
> > Besides the empirical comparison, could the authors provide an understanding on why directly fitting the map could be better than other few step generation methods based on updating samples according to the velocity field? It seems to me that the difficulty in fitting the velocity field is not less than directly updating the samples through the distilled velocity field in previous few step generation methods like diffusion distillation.
>
> Our intuition is that when directly fitting the map results in a **1-step mapping** that links the initial Gaussian noise to any point on the trajectory, evaluating a sample at any intermediate tilmestep $t_i$ then becomes as simple as evaluating the flow parametrized by the basis functions: $x_{t_i} = \psi_\theta(x_0, t=t_i)$. We believe this provides a **more compact and efficient representation** of the trajectory compared to few-step methods, which require sequential updates along the velocity field .

---

> ### Author Response · Authors · 2025-12-04
>
> > Could the flow fitting strategy be applied to other classical flow trajectories like diffusion paths in one step generation tasks like diffusion distillation? If yes, will the type of the true flow trajectory affect the choice of the basis function family and the difficulty of flow fitting?
>
> Theoretically, we rely on the **Stone–Weierstrass theorem**, which guarantees the existence of a basis function approximation on a compact domain, specifically, there exists a **polynomial basis** capable of approximating any continuous function. This result is **agnostic to the type of function being fitted**, whether diffusion-based or flow-matching-based. Therefore, we do not see any fundamental obstacle to applying the same flow fitting strategy to deterministic diffusion models such as DDIM.
> In practice, the choice of basis order and the difficulty of fitting may depend on the smoothness and complexity of the underlying trajectory. Increasing the basis order improves approximation accuracy but comes at the cost of higher computational requirements. Nonetheless, the general principle, approximating the trajectory using a **compact and expressive basis**, remains valid.

---

### Official Review · Reviewer_a4C4 · 2025-11-01

**Soundness:** 3
**Presentation:** 3
**Contribution:** 3
**Rating:** 6
**Confidence:** 5

**Summary:**

This manuscript proposes a new type of 1-step generative models called FlowFit. FlowFit approximates the flow trajectory using basis functions, and trains networks to predict the coefficients of the bases. Training is conducted by simultaneously training a flow matching model to learn the velocity field, and matching the derivative of the approximated trajectory to the learned velocity field. The proposed method shows promising results on the DiT-B architecture. Although the FID is not as strong as the SoTAs (e.g., MeanFlow DiT-B FID is 6.17 on ImageNet), the novelty of the method itself is a solid contribution that could inspire future works.

**Strengths:**

- The idea of fitting basis functions to approximate the flow trajectory is very original, and the training objective of matching the derivative of the function also differs from previous one/few-step models, such as consistency models. The fact that a network is able to predict all the coefficients of the bases in one step is very intriguing.
- Apart from the novelty, the method is also relatively simple and elegant, in contrast to SOTA consistency models which often involve adaptive losses, complex scheduling or inefficient JVP.
- The generation quality, as measured by FID, appears strong enough to outperform older generations of consistency models, such as iCT, sCT.
- The presentation is clear and focused, although adding a graphical illustration may further strengthen the clarity.

**Weaknesses:**

- The major limitation of this work is that the FID is relatively underperforming compared to SOTA consistency models. For example, on ImageNet 256x256, MeanFlow (also using DiT-B) achieves an FID of 6.17, whereas the proposed approach attains an FID of 34.4.
- Experiments are only done on DiT-B. Larger models (e.g., DiT-XL) are not tested.
- The proposed method requires training two models, one for flow matching and one for trajectory matching. This is more like online distillation rather than "end-to-end".
- Minor formatting issue: the reference format is not following the official ICLR template, which should be name + year instead of numbers.

**Questions:**

The paper shows improving performance with increasing order, but stopped at order 8. In L411, the authors explain that this is due to computational constraints. Why is maximum order limited by computation? I thought most of the computation expenses are from the neural network, and increasing the number of bases would not introduce significant overhead in training and inference (because evaluating these simple functions is usually fast)?

**Details Of Ethics Concerns:**

Potential dual submission, highly similar to this submission:
https://openreview.net/forum?id=XpOnko2c59

---

> ### Author Response · Authors · 2025-12-03
>
> Thank you for your feedback. Please see the detailed response below.
>
> > The major limitation of this work is that the FID is relatively underperforming compared to SOTA consistency models. For example, on ImageNet 256x256, MeanFlow (also using DiT-B) achieves an FID of 6.17, whereas the proposed approach attains an FID of 34.4.
>
> We would like to emphasize that these results are **not directly comparable**, primarily due to differences in the classifier-free guidance (CFG) scale used. In the case of MeanFlow, the reported FID of 6.17 corresponds to *CFG = 3*. In contrast, for a **fair comparison**, we follow Shortcut Models and use *CFG = 1.5*, which is consistent with the setup for all methods reported in Table 1 of our main paper. The impact of CFG on FID is substantial, as reported in **Table 1-(f) of the MeanFlow paper**. For instance, MeanFlow achieves an FID of 33.33 with *CFG = 1.5*, but 15.53 with *CFG = 3*. This demonstrates that the choice of CFG is a **significant hyperparameter** influencing FID, and direct comparisons without aligning CFG values can be misleading.
>
> > Experiments are only done on DiT-B. Larger models (e.g., DiT-XL) are not tested.
>
> Thanks for your suggestion. We included a comparison with larger model architectures, including DiT-XL. Here, we report the FID of 1-step inference in the case of CelebA-HQ with 400,000 training iterations.
>
> | Method |  DiT-B   |    DiT-L  |  DiT-XL|
> | ------ | ------ | ------ | ------ |
> | Ours   | 14.1   | 10.7    | 6.2   |
>
> > The proposed method requires training two models, one for flow matching and one for trajectory matching. This is more like online distillation rather than "end-to-end".
>
> When comparing with existing end-to-end 1-step diffusion and flow-matching methods, we believe that the **number of forward and backward passes per iteration** during training is what truly matters, rather than the number of networks being trained. The table below summarizes this comparison.
>
> | Method | Number of Forward  passes  |Number  of Backward passes |
> |:------:|------------------------|---------------------------|
> | Consistency Models [26, 23, 24] | 2  | 1 |
> | Shortcut Models [5]    | 3 | 1 |
> | Ours  |3 | 2 |
>
> Our method introduces a slight computational overhead during training compared to state-of-the-art Shortcut Models. However, since we train **two separate networks**, some of the forward and backward passes can be executed **in parallel**, reducing the effective sequential cost. In practice, our total compute is only marginally higher than Shortcut Models. Shortcut Models compress both the velocity predictor and the shortcut predictor into a single network $f(x_t, t, d)$, where $d = 0$ outputs the velocity and $0 < d \le 1$ outputs the shortcut prediction. This forces one architecture to represent two functions simultaneously, creating a trade-off in representational capacity and accuracy. In contrast, our method uses **two specialized networks**, each dedicated to a distinct role, and fully parallelizable during training. This design increases effective capacity without increasing sequential training cost.
>
> Finally, we believe that our formulation can also be implemented using a single unified network (similar to Shortcut Models) with an additional input flag, while still preserving our basis-function structure. We chose the two-network design because it enables parallel training and provides greater modeling flexibility.
>
> > Why is maximum order limited by computation?
>
> We use a single network to compute all basis coefficients, and each coefficient requires a different conditioning input. Since in our implementation during training the basis coefficients are computed along the batch dimension, increasing the basis order results in additional memory usage and computational overhead.
>
> > Minor formatting issue: the reference format is not following the official ICLR template, which should be name + year instead of numbers.
>
> Thank you for pointing this out. To the best of our knowledge, the current reference format we use follows the ICLR 2026 official template.

---

### Official Review · Reviewer_me4z · 2025-11-01

**Soundness:** 1
**Presentation:** 1
**Contribution:** 2
**Rating:** 2
**Confidence:** 5

**Summary:**

This paper proposes flow fitting that train two networks which are velocity flow matching and flow model that align with the velocity flow matching model. The flow trajectory model is defined in terms of basis function.

**Strengths:**

1. The model proposes to use separate network called trajectory flow to fit with training velocity model, which is new idea than the unified model like the meanflow and shortcut model.
2. The model performance beats shortcut in some setting.
3. The code implementation are provided.

**Weaknesses:**

1. The paper writing seems unprofessional with poor writing in introduction (too many short paragraph). The related works is not well-studied and lacks many recent works like [1, 2, 3].

[1]: Inductive Moment Matching - ICML

[2]: Improved Training Technique for Latent Consistency Models - ICLR

[3]: Consistency Models Made Easy - ICLR

2. The authors do not clarify why not use single deep learning model instead of both basis function and deep network to estimate coefficient.

3. The paper only compares with shortcut model in limited setting (undertraining). How about training model until converge with DiT-XL/2.

4. In ablation study, the experiment details are not provided like what dataset, training iterations and other details. The authors just simply put a table without any explanation to guide the reader.

**Questions:**

Please see the weakness

---

> ### Author Response · Authors · 2025-12-03
>
> Thank you for your feedback. Please see the detailed response below.
>
> >  The paper writing seems unprofessional with poor writing in introduction (too many short paragraph).
>
> Thanks for pointing this out. We did our best to improve the writing and in particular the introduction.
>
> > The related works is not well-studied and lacks many recent works like [1, 2, 3]
>
> We added a discussion with the mentioned works [1, 2, 3] in our revision as well as a quantitative comparison using the same batch size, cfg scale, network architecture and compute budget.
>
> | Method              | CelebA-HQ (DiT-B/2) | ImageNet (DiT-B/2) |
> | ------------------- | -------------------- |-------------------- |
> | IMM [1]            | 19.5                  |41.4                  |
> | sLCT [2]            | 18.8                |39.9              |
> | ECT [3] | 20.7                  | 40.6             |
> | Ours            | **14.1**              |**34.4**                  |
>
> > The authors do not clarify why not use single deep learning model instead of both basis function and deep network to estimate coefficient.
>
> One of our main motivations for using a basis model $\psi_\theta = c_1(\theta ) \cdot t + c_2(\theta ) \cdot t^2 + .. +  c_n(\theta ) \cdot t^n$ rather than a single network $\psi_\theta(x, t)$ is to **decouple time and space** in modeling the flow map. This allows us to obtain the **exact** derivative with respect to $t$ analytically **“for free”**, leading to improved computational efficiency. Since our method relies on matching the velocity field to the derivative of the flow with respect to $t$, a basis formulation enables this derivative to be computed analytically, for example, $\frac{d \psi_\theta}{dt} = c_1(\theta ) + 2 c_2(\theta ) \cdot t + .. +  n c_n(\theta ) \cdot t^{n-1}$, and at no additional computational cost. In contrast, using a single network would require backpropagation through the network with respect to $t$ (in addition to backpropagating through $\theta$), making training significantly more expensive.
>
> > The paper only compares with shortcut model in limited setting (undertraining). How about training model until converge with DiT-XL/2.
>
> Thank you for the suggestion. We added a comparison using a larger architecture (DiT-XL/2) and increased the training iterations (doubling those used in Table 1 of the main paper) using the same batch size, cfg scale and compute budget. The results  are reported below:
>
> | Method              | CelebA-HQ (DiT-XL/2) | ImageNet (DiT-XL/2) |
> | ------------------- | -------------------- |-------------------- |
> | iCT [23]            | 9.4                  |22.1                  |
> | sCT [14]            | 8.2                  |20.5                 |
> | Shortcut Models [5] | 10.7                  | 23.6                  |
> | Ours           | **5.2**              |**16.6**                  |
>
> Our method continues to outperform competing approaches even when both the **model capacity** and the **training budget** are increased.
>
> > In ablation study, the experiment details are not provided like what dataset, training iterations and other details. The authors just simply put a table without any explanation to guide the reader.
>
> Thank you for pointing this out. We have added more details to the ablation study, including the dataset, training iterations, and additional explanations to guide the reader through the table.
>
> -------
> [1] Inductive Moment Matching (IMM)- ICML
>
> [2] Improved Training Technique for Latent Consistency Models (sLCT) - ICLR
>
> [3] Consistency Models Made Easy (ECT) - ICLR

---

### Meta-Review · Area_Chair_MUZu · 2026-01-12

**Summary:**

### Reviewer me4z
1. Paper writing. The paper writing is unprofessional and lacks many recent works. Lack of training details.
    * **Author replies**: The paper has been revised the paper to include more related works and training details.
    * **AC comment**: I think the concerns are well addressed.
2. Why use basis function instead of a single network?
    * **Author replies**: The authors point out that the motivation is to decouple time and space
    in modeling the flow map so that the derivative w.r.t. $t$ can be analytically given.
    * **AC comment**: I think the concerns are well addressed.
3. Missing comparisons on larger models such as DiT-XL/2
    * **Author replies**: The authors added experiments on DiT-XL/2 and show that it achieves better performance.
    * **AC comment**: I think the concerns are well addressed.



### Reviewer a4C4
1. FID of the proposed method underperforms MeanFlow.
    * **Author replies**: The paper claims that the results are not directly comparable since MeanFlow uses
    a different CFG scale.
    * **AC comment**: While the authors' replies are reasonable, I think the paper still needs to include a
    comparison with MeanFlow for completeness.
2. Missing comparisons on larger models such as DiT-XL/2
    * See Reviewer me4z 3
3. The paper is more like online distillation rather than "end-to-end"
    * **Author replies**: The authors argue that it's not the number of networks but the number
    of fwd/bwd passes that matter, and also emphasize the benefits of two separate networks.
    * **AC comment**: I think the authors do not fully address the concern here. I think the reviewer
    here means the paper uses two separate networks, and while these two networks are trained simultaneously
    in an end-to-end manner, the velocity network is more used to provide a supervision signal to the basis
    network. Therefore, it is totally possible to train the velocity network first, and then do distillation, while
    what the paper currently does more resembles "online distillation". The paper has not demonstrated the benefits
    of training two networks simultaneously.
4. Ethical concerns. The paper has high similarity to another submission to ICLR.
    * **AC comment**: The ethical concern has been reported to the Ethics Chairs.

### Reviewer YPvN
1. Add more discussions on the choice of basis functions and their capacity.  Whether the proposed method can scale up to more complex generation tasks
    * **Author replies**: The paper turns to Stone–Weierstrass theorem to explain the capacity of the basis function.
    * **AC comment**: I think the concern remains unresolved. It's not clear when it comes to more
    complex generation tasks, whether the method would require a larger number of bases, and how many bases
    are needed. A larger number of bases will also impact the training/inference time.
2. Unfair comparison. While the proposed method performs 1-step, its complexity is much higher than
standard flow matching or consistency model.
    * **Author replies**: The paper lists the inference time of the proposed method and baseline methods.
    The paper argues that the proposed method can make use multi-process to achieve similar inference time
    with baseline methods when using basis order 8.
    * **AC comment**: I think the concern remains unresolved. While the proposed method performs 1-step inference,
    it requires multiple network evaluations that equal the basis order. The authors argue that these networks
    evaluations can be done in parallel via multi-process to achieve a similar inference time as the baseline,
    However, it means that the proposed method will require more computing than baseline methods, and in fact the
    baseline methods can also make use of extra compute for acceleration (such as tensor/sequence parallel).


### Reviewer vFLX
1. The method is overall naive.
    * **Author replies**: The authors make further clarifications on the novelty.
    * **AC comment**: I think the reviewer's opinion is not well-grounded.
2. The computation of the method is doubled due to the need for two separate networks.
    * **Author replies**: The authors argue that while the proposed method requires more fwd/bwd passes,
    some of them can be done in parallel, thus reducing the effective sequential cost.
    * **AC comment**: The concern remain outstanding. While some fwd/bwd passes can be done in parallel,
    it still requires extra compute and can make the proposed method slower than baseline methods, given
    the same compute.

**Reviewer Concerns:**

Some of the concerns remain outstanding (see above for details):
1. Unfair comparison due to the need for more compute during inference
2. The paper is more like online distillation rather than "end-to-end."
3. More training compute is needed due to training two separate networks
4. Scale to more complex generation tasks.

In addition, the ethical concerns on dual submission also remain unresolved.

**Reviewer Scores:**

Reviewer me4z: improve from 2 to 4

Reviewer a4C4: decrease from 6 to 4 or 2

Reviewer YPvN: maintain 4

Reviewer vFLX: maintain 4

---

### Decision · Program_Chairs · 2026-01-26

Reject